# Efficient Model Assisted Probability of Detection Estimations in Eddy Current NDT with ACA-SVD Based Forward Solver

**DOI:** 10.3390/s22197625

**Published:** 2022-10-08

**Authors:** Yang Bao, Minxuan Xu, Jiahao Qiu, Jiming Song

**Affiliations:** 1College of Electronic and Optical Engineering, Nanjing University of Posts and Telecommunications, Nanjing 210023, China; 2State Key Laboratory of Millimeter Waves, Southeast University, Nanjing 210096, China; 3Center for Nondestructive Evaluation, Department of Electrical and Computer Engineering, Iowa State University, Ames, IA 50011-2042, USA

**Keywords:** model assisted probability of detection (MAPoD), eddy current, nondestructive testing (NDT), singular value decomposition (SVD), adaptive cross approximation (ACA), boundary element method (BEM)

## Abstract

Model assisted probability of detection (MAPoD) is crucial for quantifying the inspection capability of a nondestructive testing (NDT) system which uses the coil or probe to sense the size and location of the cracks. Unfortunately, it may be computationally intensive for the simulation models. To improve the efficiency of the MAPoD, in this article, an efficient 3D eddy current nondestructive evaluation (ECNDE) forward solver is proposed to make estimations for PoD study. It is the first time that singular value decomposition (SVD) is used as the recompression technique to improve the overall performance of the adaptive cross approximation (ACA) algorithm-based boundary element method (BEM) ECNDE forward solver for implementation of PoD. Both the robustness and efficiency of the proposed solver are demonstrated and testified by comparing the predicted impedance variations of the coil with analytical, semi-analytical and experimental benchmarks. Calculation of PoD curves assisted by the proposed simulation model is performed on a finite thickness plate with a rectangular surface flaw. The features, which are the maximum impedance variations of the coil for various flaw lengths, are obtained entirely by the proposed model with selection of the liftoff distance as the uncertain parameter in a Gaussian distribution. The results show that the proposed ACA-SVD based BEM fast ECNDE forward solver is an excellent simulation model to make estimations for MAPoD study.

## 1. Introduction

Nondestructive testing/evaluation (NDT/E) is a popular research area which aims at detecting the flaws of materials and characterizing their discontinuities without destroying their serviceability [1,2,3]. This non-invasive method has many applications in aerospace, civil engineering, nuclear industry and so on [4,5,6]. The materials are considered to be harmless if the dimension of the flaw is trivial, while it is necessary to replace those materials with a significant flaw whose dimension exceeds a certain value [7,8]. NDT/E plays an important role in evaluating or testing materials to see whether their flaws or cracks affect the integrity of the structure. Various factors may introduce uncertainty into the reliability of the NDT system and this can be quantified by the probability of detection (PoD) [9,10]. PoD(*a*) determines the probability that a flaw with a size of *a* will be detected, considering of the uncertainties due to human, device or other factors [11]. The PoD calculation was initially developed and conducted using experiments only. However, it takes much time to accurately calculate the parameters of a reliable statistical model with the large amount of data from a sufficient number of flaws by experiments [12]. Thus, simulation models are proposed to deal with the required large data set.

The simulation model can make accurate predictions efficiently for the NDT system responses as part of the data needed while the rest of the data is obtained by experiment. This approach is the model assisted PoD (MAPoD) which mitigates the extensive amount of empirical data required in PoD study [13]. A wide range of physics-based numerical simulation models for eddy current NDT systems have been proposed [14,15,16,17,18,19,20,21,22]. The finite element method (FEM), which solves the differential equations, draws extensive attentions in MAPoD study because it is easier to implement than the boundary element method (BEM) or volume element method (VEM), which solve the integral equations [14,15,16,17]. However, it discretizes the whole solution domain with volume meshes which need large computational resources in both FEM and VEM [23]. BEM has the merits that only the surface meshes of the considered domain need to be discretized which leads to a reduction in the number of unknowns and has been applied to compute the flaw responses in ECNDE forward problems and MAPoD study [18,19,20,21,22].

In BEM, both the memory requirement and computational time grow in proportion to $O\left(N^{2}\right)$ with iterative solvers. As the size of the object under detection becomes larger, to maintain good accuracy, the required number of surface elements is increased which results in low computational efficiency [24]. To alleviate this problem, fast algorithms have since been proposed. To name a few, there are the multilevel fast multipole algorithm (MLFMA) [25,26], the adaptive cross approximation algorithm (ACA) [27,28,29,30], the *H* matrix algorithm [31,32,33,34] and so on. The ACA algorithm is purely algebraic and kernel independent which makes it one of the best algorithms to compress low rank matrices [27]. ACA is also a versatile tool and implementable to existing codes easily. In the ACA algorithm, the low rank matrix **Z** would be approximated by the multiplication of the **U** and **V** matrices. The columns of matrices **U** and **V** are usually not orthogonal and may contain redundancies which could be removed with the algebraic recompression technique that is called the singular value decomposition (SVD) [35].

In this article, unlike the conventional eddy current BEM based MAPoD, the PoD study is assisted by the novel ACA-SVD accelerated BEM forward solver. The required flaw features are obtained by the proposed physics-based numerical model which improves the overall performance of the BEM based ECNDE solver and has extra savings in both memory requirement and CPU time compared to an ACA based solver. To find the low rank matrices, the object under detection is enclosed in a cube and partitioned into smaller blocks until each one contains certain number of unknowns. Then, the block pairs are categorized into near and far blocks, based on the distance between blocks. The near block pairs are fully computed by BEM while the far block ones are approximated by the ACA-SVD algorithm. The PoD curves are performed and assisted by the proposed forward solver, which is validated by comparing the predicted flaw responses in benchmarks, on a finite thickness plate with a rectangular surface flaw. The flaw features of various lengths are obtained by selecting the liftoff distance as the uncertain parameter in Gaussian distribution. It shows that the proposed ACA-SVD based BEM ECNDE forward solver is an efficient simulation model to make the large number of predictions required in MAPoD study.

The procedure of the ACA-SVD based BEM forward solver for MAPoD study of eddy current problems is shown in Figure 1. With the stated problem, the uncertain parameters are determined and the parameter settings are sent to ACA-SVD based BEM forward solver to make predictions. For PoD study, the required large amounts of predicted data are generated by the forward solver. Then the PoD study is conducted, which includes the regression analysis, and the PoD curve is drawn.

## 2. ACA-SVD Based BEM Forward Solver for Eddy Current NDE

### 2.1. BEM Model

In the PoD study, due to the fact that to obtain the large amount of data experimentally takes plenty of time, the efficient physics-based numerical model is needed. Here this model is implemented by BEM. The formulation selected is the Stratton–Chu formulation which contains both the tangential and normal components of the surface electromagnetic fields explicitly [36]
(1)$$E(r)=E^{inc}(r)+\oint_{S}\left\{\left[\hat{n}\cdot E(r^{\prime })\right]\nabla^{\prime }G(r,r^{\prime })+\left[\hat{n}\times E(r^{\prime })\right]\times \nabla^{\prime }G(r,r^{\prime })-j\omega \mu G(r,r^{\prime })\left[\hat{n}\times H(r^{\prime })\right]\right\}dS^{\prime },$$
(2)$$H(r)=H^{inc}(r)+\oint_{S}\left\{j\omega \varepsilon G(r,r^{\prime })\left[\hat{n}\times E(r^{\prime })\right]+\left[\hat{n}\cdot H(r^{\prime })\right]\nabla^{\prime }G(r,r^{\prime })+\left[\hat{n}\times H(r^{\prime })\right]\times \nabla^{\prime }G(r,r^{\prime })\right\}dS^{\prime },$$
where $E^{inc}$ and $H^{inc}$ are the incident electromagnetic fields. r and $r^{\prime }\in S$ are the field and source points in the domain of interest, respectively. $\nabla^{\prime }$ is the gradient with respect to $r^{\prime }$, $\hat{n}$ is the unit normal direction pointing towards the solution domain and $G(r,\;r^{\prime })$ is the Green function.

Once the formulation is selected, the basic steps of BEM would be applied. Although the details of BEM based ECNDE solver have been given elsewhere, for the sake of completeness, here we make brief reviews [37,38]. In the ECT operating frequency range, the displacement currents in the metal are trivial and can be ignored as compared with the conducting currents. Thus, the Stratton–Chu formulation can be approximated with the low frequency and high conductivity approximations which have already been validated in [38]. The normal component of the magnetic field, and the equivalent electric and magnetic surface currents are expanded by the pulse basis function and the RWG vector basis function [39]. The discretized impedance matrix then reads [37]
(3)$$\left[\begin{array}{ccc} 0.5T-K_{1}^{\times } & 0 & R_{1}^{\times }\\ j\mu_{2}/\mu_{1}L_{2}^{\times } & 0.5T+K_{2}^{\times } & 0\\ \mu_{2}/\mu_{1}K_{2}^{n} & -jk_{2}^{2}L_{2}^{n} & 0.5D-R_{2}^{n} \end{array}\right],$$
where subscript ***l*** of K, L and R operators is 1 or 2 and stands for air region or metal region; the superscripts × and n denote the cross or dot product with $\hat{n}$, and the K, L and R operators are [37]
(4)$$K_{l}\left(X\right)=P.V.\;\oint_{S}\nabla G_{l}\left(r,\;r^{\prime }\right)\times X\left(r^{\prime }\right)\;dS^{\prime },$$
(5)$$L_{l}\left(X\right)=\oint_{S}G_{l}\left(r,\;r^{\prime }\right)X\left(r^{\prime }\right)\;dS^{\prime },$$
(6)$$R_{l}\left(X_{n}\right)=P.V.\;\oint_{S}\nabla G_{l}\left(r,\;r^{\prime }\right)X_{n}\left(r^{\prime }\right)\;dS^{\prime },$$
where *P.V*. indicates that the integrals are treated as principal values.

The total number of unknowns for the BEM model as shown in (3) is $2N_{e}+N_{p}$, where $N_{e}$ is the number of edges and $N_{p}$ is the number of triangles. The system of equations is solved by iterative solvers, such as the GMRES [40]. The CPU time complexity of the specific Krylov method can be estimated as $aN_{iter}O\left(N^{2}\right)$, where *a* is the number of matrix vector multiplications per each iteration, and $N_{iter}$ is the number of iterations required to converge to a given relative residual [40].

### 2.2. ACA-SVD Algorithm

The impedance matrix obtained from BEM of the Stratton–Chu formulation with low frequency and high conductivity approximations is not rank deficient. Due to the nature of the Green function, in the impedance matrix, there are many rank deficient submatrices. To find them, the basis functions in the geometry are grouped into sub-blocks which splits the whole impedance matrix into submatrices. Based on the distance between blocks, there are near block pairs and far block ones. The near block pairs’ interactions are computed and saved directly, while, the numerical rank deficient submatrices related to the far block pairs are compressed by the ACA-SVD algorithm.

Suppose matrix $Z^{m\times n}$ represents the interaction of two well-separated blocks. In the ACA algorithm, the matrix ${\tilde{Z}}^{m\times n}$ is the multiplication of matrices $U^{m\times r}$ and $V^{r\times n}$ to approximate the matrix $Z^{m\times n}$
(7)$${\tilde{Z}}^{m\times n}=U^{m\times r}V^{r\times n},$$

Because r≪min(m,n), instead of computing and storing m×n entries, only r×(m+n) entries are needed. This shows that only a few rows and columns of matrix **Z** can represent it with a desired accuracy.

The basic procedures of ACA algorithm can be found in [27,28,29,30]. The first step is to pivot the row index arbitrarily to select the first row $V_{1}=Z\left(i_{1},:\right)$. Then find the maximum value of $V_{1}$ as $V_{1\max}$ and its position $j_{1}$. In the second step, the first column can be found $U_{1}=Z\left(:,j_{1}\right)/V_{1\max}$. Find the maximum value’s position in $U_{1}$ as $i_{2}$. Then $V_{2}$ is $Z\left(i_{2},:\right)-U_{1}\left(i_{2}\right)V_{1}$. As in the first step, find the maximum value and its position in $V_{2}$ as $V_{2\max}$ and $j_{2}$. $U_{2}$ can be achieved by $\left[Z\left(:,j_{2}\right)-U_{1}V_{1}\left(j_{2}\right)\right]/V_{2\max}$. The third step, for the *k*th iteration, follows the steps mentioned with $V_{k}=Z\left(i_{k},:\right)-\sum_{j=1}^{k-1}U_{j}\left(i_{k}\right)V_{j}$ and $U_{k}=\left[Z\left(:,j_{k}\right)-\sum_{j=1}^{k-1}V_{j}\left(j_{k}\right)U_{j}\right]/V_{k\max}$. The tolerance is τ and the stopping criterion is
(8)$$\left(\left|U_{k}\right|\left|V_{k}\right|\right)/\left(\left|U_{1}\right|\left|V_{1}\right|\right)\leq \tau ,$$
where |·| refers to the Euclidean norm [41].

There may contain redundancies in U and V matrices because their columns are usually not orthogonal. The redundancies can be removed by the recompression technique which is called the singular value decomposition optimization [35]. With the QR decompositions $U=Q_{U}^{}R_{U}^{}$ and $V^{T}=Q_{V}^{}R_{V}^{}$, the products of $R_{U}$ and $R_{V}^{T}$ matrices can be decomposed by SVD as
(9)$$R_{U}R_{V}^{T}=\hat{U}\hat{\Sigma }\hat{V},$$

The original matrix **Z** is approximated as
(10)$$Z=\tilde{U}\times \tilde{V},$$
where $\tilde{U}=Q_{U}\hat{U}\hat{\Sigma }$ and $\tilde{V}={\hat{V}Q}_{V}^{T}$. The SVD works as the post compression technique to reduce the required storage in the ACA algorithm.

### 2.3. Validation of the ACA-SVD Based BEM Soler

In this section, the accuracy and efficiency of the ACA-SVD based BEM forward solver is validated by comparing the predicted impedance variations with the values from analytical, semi-analytical, and experimental results. All computations are done in a double precision on an AMD Workstation with a clock speed of 3.7 GHz and 256 GB of RAM.

#### 2.3.1. Coil with a Finite Cross Section Placed above the Conducting Plate

The first benchmark case presented is placing a coil with a finite cross section above a conducting plate. The information on coils and conducting plate can be found in Table 1 and Table 2 and the *n*-turn coil with rectangular cross section is shown in Figure 2 [38,42].

For placing coil C27 above conductive plate B2, the operating frequency is 20 kHz. The solution domain is 80 mm by 80 mm. Total number of unknowns is 15,176 with the edge length of the mesh 2.14 mm. The impedance variations predicted by the proposed ACA-SVD based BEM forward solver are compared with analytical, semi-analytical methods and experimental values as shown in Table 3. In the ACA-SVD solver, the tolerance $\tau =10^{-1}$ and truncated error of SVD $e=10^{-1}$. Good agreements of the impedance variations among the ACA-SVD, ACA, BEM, analytical, semi-analytical methods and experiments can be observed. The relative differences among them are smaller than 1% in both real and imaginary parts of the impedance variations. In the memory requirement of the far block interactions, ACA-SVD based BEM ($\tau =10^{-3}$, truncated error$=10^{-1}$) costs 75.1% less than ACA based BEM ($\tau =10^{-3}$). As to the overall performance, the ACA-SVD based BEM only needs 4.15% memory and 9.01% CPU time per iteration of BEM forward solver.

The memory requirement of far block interactions and CPU time per iteration of ACA-SVD based BEM forward solver with ACA tolerances $\tau =10^{-1},10^{-2},10^{-3}$ and SVD truncated errors $e=10^{-1},10^{-2},10^{-3},10^{-4}$ are shown in Figure 3a,b, respectively. For accuracy, the predicted impedance variations with the parameter settings in Figure 3 all agree well with those from other methods. For performance, it can be found that both the memory requirements of far block interactions and CPU time per iteration decrease as the truncated error of SVD increases, while maintaining the tolerance of the ACA algorithm. With ACA tolerance $\tau =10^{-3}$, ACA-SVD with truncated error $e=10^{-1}$ has 47.7%, 63.3%, 71.7%, and 74.9% savings compared to ACA-SVD with $e=10^{-2},10^{-3},10^{-4}$ and pure ACA algorithm in the memory requirements for far block interactions. Those savings are 16.8%, 30.1%, 38.1%, 41.0% for the CPU time per iteration. For the ACA-SVD solver with ACA tolerance $\tau =10^{-1},10^{-2},10^{-3}$ and SVD truncated error $e=10^{-1}$, it has 52.8%, 66.9%, and 74.8% savings in memory requirement of far block interactions and 21.0%, 35.1%, and 40.8% savings in the CPU time per iteration compared to the pure ACA algorithm. It can be easily concluded that the proposed ACA-SVD based BEM solver has extra savings and improves the overall performance of the pure ACA algorithm-based BEM forward solver.

As for placing coil C5 above conductive plate B1 which operates at 850 Hz, the solution domain truncates to a square with the side length 120 mm. The edge length of the mesh is 3.21 mm which results in the number of unknowns being 15,024. The impedance variations predicted by the ACA-SVD based BEM solver (with the ACA tolerance $\tau =10^{-1}$ and truncated error of SVD $e=10^{-1}$) are compared with analytical, semi-analytical methods and experiments as shown in Table 4.

Again, in both real and imaginary parts of the impedance variations, good agreements among the ACA-SVD, ACA, BEM, analytical, semi-analytical methods and experiments can be observed with the relative differences being smaller than 1%. In the memory requirement of the far block interactions, ACA-SVD based BEM ($\tau =10^{-3}$,truncated error $=10^{-1}$) costs 75.3% less than ACA based BEM ($\tau =10^{-3}$). As to the overall performance, the ACA-SVD based BEM only needs 5.58% memory and 7.16% CPU time per iteration of BEM forward solver. The memory requirement of far block interactions and CPU time per iteration of ACA-SVD based BEM forward solver with ACA tolerances $\tau =10^{-1},10^{-2},10^{-3}$ and SVD truncated errors $e=10^{-1},10^{-2},10^{-3},10^{-4}$ are shown in Figure 4a,b, respectively. All the impedance variations predicted by the mentioned ACA tolerances and SVD truncated errors agree well with the one by BEM such that the relative differences are smaller than 1%.

It can be easily seen that ACA-SVD makes extra memory and CPU time savings in the ACA algorithm: with ACA tolerance $\tau =10^{-3}$, ACA-SVD with truncated error $e=10^{-1}$ has 49.3%, 64.7%, 73.4%, and 75.3% savings compared to ACA-SVD with $e=10^{-2},10^{-3},10^{-4}$ and pure ACA algorithm in the memory requirements for far block interactions. Those savings are 25.6%, 39.1%, 51.4%, 54.1% for CPU time per iteration. For ACA-SVD solver with ACA tolerance $\tau =10^{-1},10^{-2},10^{-3}$ and SVD truncated error $e=10^{-1}$, it has 51.2%, 67%, and 75.5% savings in memory requirement of far block interactions and 20%, 41.2%, and 54.1% savings in CPU time per iteration compared to the pure ACA algorithm.

#### 2.3.2. Coil with a Finite Cross Section Placed above the Thick Plate with a Surface Slot

The second testing case is TEAM workshop benchmark problem 15 which is placing the coil with a finite cross section above a thick plate with a surface slot [44]. The coil, which operates at 900 Hz, with the inner radius 6.15 mm, outer radius 12.4 mm, and 3790 turns is scanning parallel to the *x* axis along the length of a rectangular slot. The conducting plate has thickness 6.15 mm and conductivity 30.6 MS/m. The liftoff-distance of the coil is 0.88 mm. The surface slot has length 12.6 mm, depth 5 mm and width 0.28 mm. The solution domain is a square with the side length 10 times of the coil outer radius. The edge length of the mesh is 1.5 mm with the number of unknowns 110,000. ACA-SVD based BEM solver is applied to predict the impedance variations compared with the experiment results [44] as shown in Figure 5. The ACA tolerance is $\tau =10^{-1}$ and SVD truncated error is $e=10^{-1}$. Excellent agreements between the ACA-SVD based BEM solver and the experiments can be found in Figure 5.

## 3. PoD Analysis

Having demonstrated the robustness and efficiency of the ACA-SVD based BEM forward solver, we apply the proposed solver to make estimations for the PoD study.

The PoD curve relates the probability of detecting a flaw with certain sizes. PoD calculations can be performed with two statistical methodologies: the “$\hat{a}$ vs. a” regression analysis and “hit/miss” analysis [45]. For “hit/miss” analysis, the NDT system responses larger than the threshold are regarded as 1 (“hit”), and others as 0 (“missed”). The PoD of detecting the flaw can be estimated with this binary information. In the “$\hat{a}$ vs. a” regression analysis, the flaw response ($\hat{a}$) is assumed to be proportional to the flaw size (*a*). With the defined threshold ${\hat{a}}_{th}$, the PoD can be calculated [11]
(11)$$\mathrm{PoD}\left(a\right)=\Phi \left[\frac{\ln\left(a\right)-\mu }{\sigma }\right],$$
where Φ denotes the normal cumulative distribution function; the mean μ and standard deviation σ both on log scale can be represented as
(12)$$\mu =\frac{\ln\left({\hat{a}}_{th}\right)-\beta_{0}}{\beta_{1}},$$
(13)$$\sigma =\frac{\sigma_{r}}{\beta_{1}},$$
where $\beta_{0}$, $\beta_{1}$ and $\sigma_{r}$ can be estimated by the maximum likelihood method [46].

For the PoD analysis, a finite thickness plate with a rectangular surface flaw is studied. The flaw lengths range from 0.1 mm to 1 mm in steps of 0.1 mm, with the depth 5 mm and width 0.28 mm. The thickness of the conducting plate is 12.22 mm with the conductivity 30.6 MS/m. The coil with a finite cross section has the inner radius 9.34 mm, outer radius 18.4 mm, and 408 turns. The flaw features, which are the maximum impedance variations of the coil, are predicted at 7000 Hz. To account for the uncertainty due to the imperfection in inspections, lift-off variations are taken as the uncertain parameters. The lift-off variations are assumed to be normal distributed with mean 2 mm and variance 0.5 mm. For a given flaw length, 45 lift-off distances are generated and the corresponding simulations are carried out.

To generate the surface meshes, the mesh sizes are from 0.8 mm to 1.5 mm depending on the flaw lengths. The solution domain is truncated into a square with a side of 10 times of the coil outer radius which makes the average number of unknowns around 100,000. The ACA tolerance $\tau =10^{-1}$, and SVD truncated error $e=10^{-1}$. The maximal amplitude of the flaw responses appears at a distance of 13 mm on the *x* axis away from the center of the flaw. The predicted amplitudes of the impedance variations are shown in Figure 6 for “$\hat{a}$ vs. a” regression analysis and PoD generation. The “$\hat{a}$ vs. a” regression analysis and PoD curve with a lower 95% confidence bound are shown in Figure 7. The threshold value is 6.52 mΩ. In the regression model, the intercept term $\beta_{0}$ is −4.05 and the slope $\beta_{1}$ is 1.66. The three key PoD metrices $a_{50}$, $a_{90}$ and $a_{90\_95}$ are 0.555 mm, 0.641 mm, and 0.650 mm, respectively.

## 4. Conclusions

In this article, a novel ACA-SVD based BEM eddy 3D current NDE forward solver is presented to assist the PoD study. Both the accuracy and efficiency of the proposed solver are validated through comparing the predicted impedance variations with other methods as benchmarks. In this application, the PoD calculation is carried out for a rectangular flaw located at the top of the thick conducting plate. The flaw features, which are the maximum variations in coil impedance, have been obtained with the proposed solver for “$\hat{a}$ vs. a” regression analysis and to draw the PoD curve.

In future work, for the PoD study including multiple uncertain parameters, such as the flaw length, flaw height, flaw width, probe location and so on, large amounts of flaw responses will need to be calculated, which will be beyond the capability of the fast solver. Thus, the metamodel or a surrogate model is required to alleviate the huge computational burden, which could be a good extension of our current work.

## Figures and Tables

**Figure 1 sensors-22-07625-f001:**
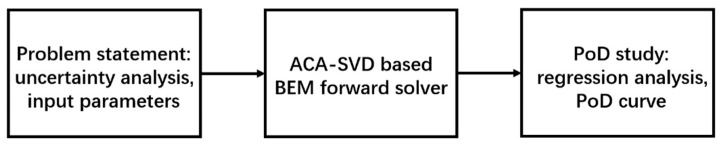
The procedure of the proposed fast forward solver based MAPoD study.

**Figure 2 sensors-22-07625-f002:**
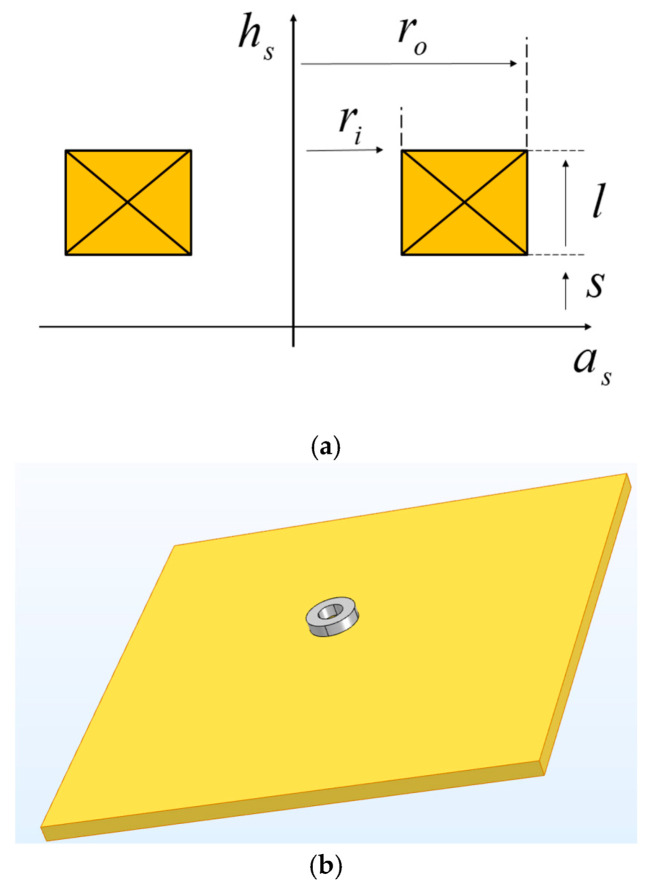
(**a**) *n*-turn coil with a rectangular cross section, with inner radius $r_{i}$, outer radius $r_{o}$, a thickness *l*, and lift-off distance *S*. (**b**) placing the coil with rectangular cross section above the object under detection.

**Figure 3 sensors-22-07625-f003:**
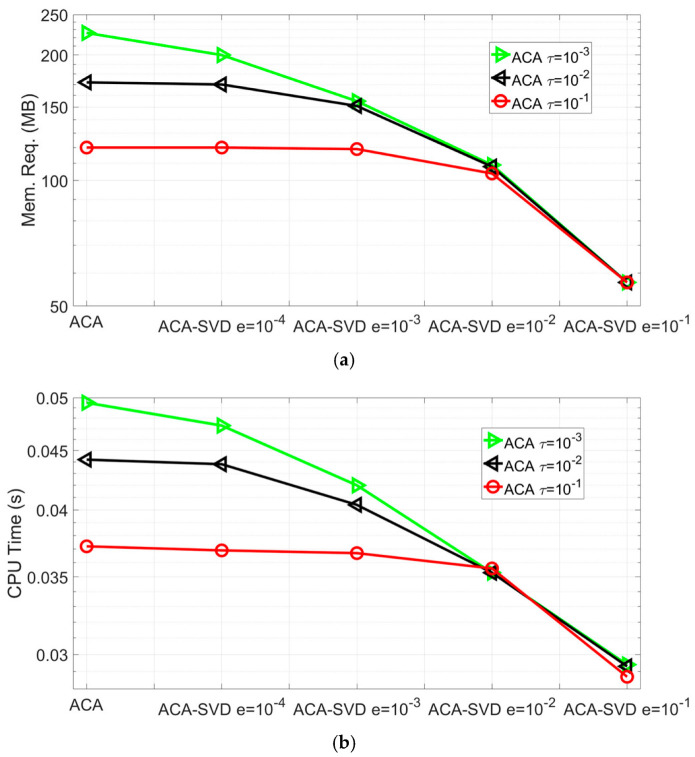
Performance of ACA-SVD based BEM forward solver for predicting the impedance variations of placing coil C27 above conducting plate B2 with ACA tolerance $\tau =10^{-1},10^{-2},10^{-3}$ and SVD truncated error $e=10^{-1},10^{-2},10^{-3},10^{-4}$. (**a**) The memory requirements of the far block interactions. (**b**) CPU time per iteration.

**Figure 4 sensors-22-07625-f004:**
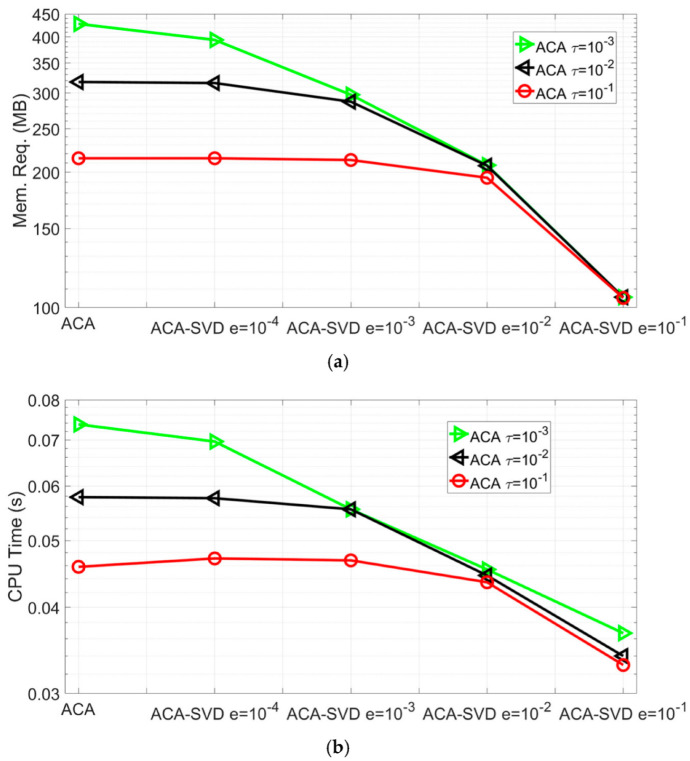
Performance of ACA-SVD based BEM forward solver for predicting the impedance variations of placing coil C5 above conducting plate B1 with ACA tolerance $\tau =10^{-1},10^{-2},10^{-3}$ and SVD truncated error $e=10^{-1},10^{-2},10^{-3},10^{-4}$. (**a**) The memory requirements of the far block interactions. (**b**) CPU time per iteration.

**Figure 5 sensors-22-07625-f005:**
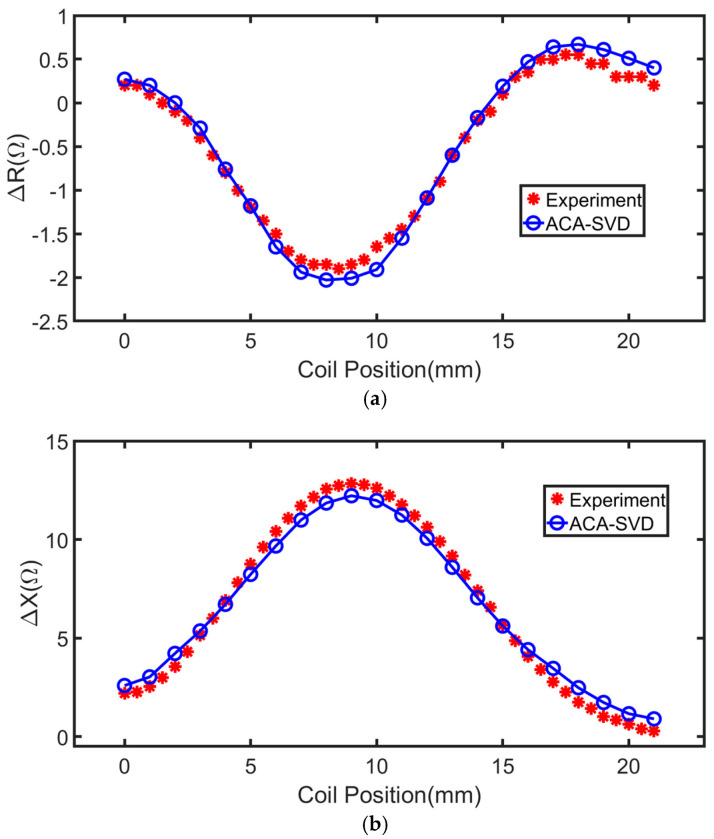
Impedance variations of scanning the coil with a finite cross section above the conducting plate with a surface slot achieved by the ACA-SVD based BEM solver and experiments, respectively. (**a**) Resistance variations. (**b**) Reactance variations.

**Figure 6 sensors-22-07625-f006:**
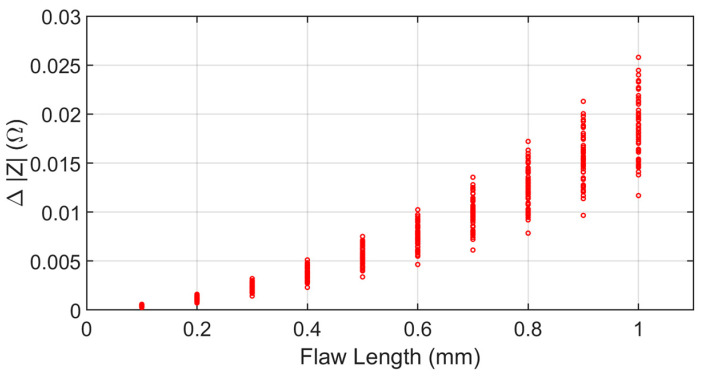
ECT responses simulated by the ACA-SVD based BEM forward solver. The amplitudes of the impedance variations are plotted against the flaw length. The flaw lengths range from 0.1 mm to 1 mm in steps of 0.1 mm. For a given flaw length, 45 predictions are carried out when taking the lift-off distance as the uncertain parameter in a Gaussian distribution.

**Figure 7 sensors-22-07625-f007:**
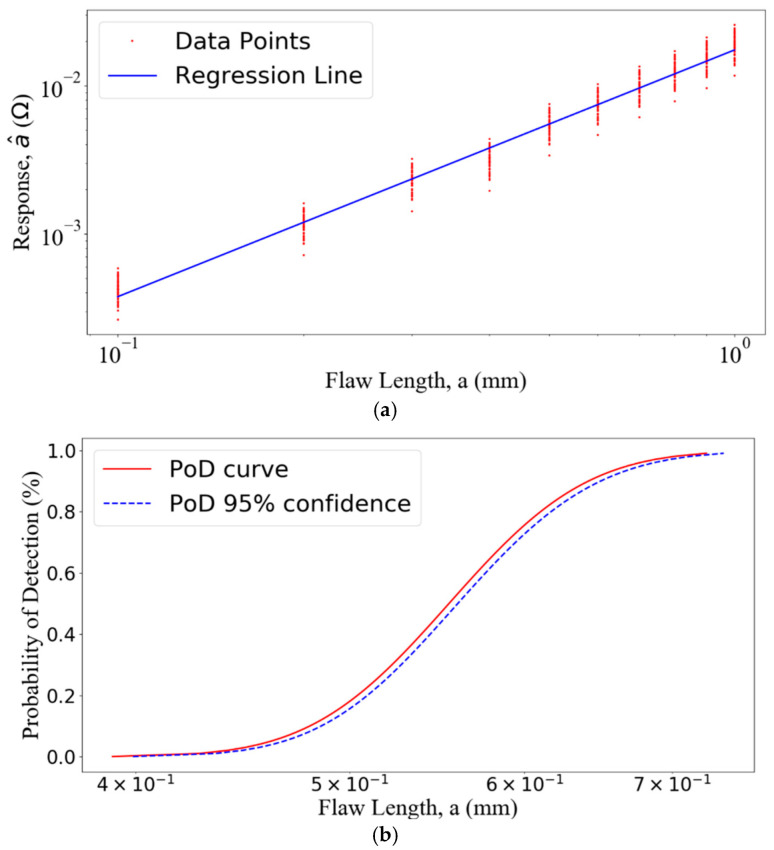
(**a**) The “$\hat{a}$ vs. a” regression analysis and (**b**) PoD curve with a lower 95% confidence bound. The three key PoD metrices $a_{50}$, $a_{90}$, and $a_{90\_95}$ are 0.555 mm, 0.641 mm, and 0.650 mm, respectively.

**Table 1 sensors-22-07625-t001:** Information on the coils with finite cross sections.

Coil Parameters	C5	C27
Inner Radius (mm)	9.33	7.04
Outer Radius (mm)	18.04	12.4
Liftoff Distance (mm)	3.32	3.43
Thickness (mm)	10.05	5.04
Number of Turns	1910	556

**Table 2 sensors-22-07625-t002:** Information on the conducting plate.

Coil Parameters	B1	B2
Conductivity (MS/m)	25.5	21.8
Thickness (mm)	140	65

**Table 3 sensors-22-07625-t003:** Impedance variations predicted by ACA-SVD based BEM, analytical, semi-analytical methods and experiments for placing coil C27 above conducting plate B2.

Method	Impedance Variation (Ω)
ACA-SVD ($\tau =10^{-1}$,$e=10^{-1}$)	12.723−124.986j
ACA ($\tau =10^{-1}$)	12.730−124.981j
BEM [30]	12.734−124.98j
Experiment [42]	12.650−125.1j
Theodoulidis and Bowler [42]	12.801−125.329j
Dodd and Deeds [43]	12.801−125.388j

**Table 4 sensors-22-07625-t004:** Impedance variations predicted by ACA-SVD based BEM, analytical, semi-analytical methods and experiments for placing coil C5 above conducting plate B1.

Method	Impedance Variation (Ω)
ACA-SVD ($\tau =10^{-1}$,$e=10^{-1}$)	22.1521−70.4310j
ACA ($\tau =10^{-1}$)	22.1501−70.4242j
BEM [30]	22.1529−70.4094j
Experiment [42]	22.00−70.5j
Theodoulidis and Bowler [42]	22.25−70.45j
Dodd and Deeds [43]	22.00−70.49j

## Data Availability

Not applicable.

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
