# Peer review of "Efficient Model Assisted Probability of Detection Estimations in Eddy Current NDT with ACA-SVD Based Forward Solver"

_sensors, 2022, doi:10.3390/s22197625_

Round 1

Reviewer 1 Report

This paper proposed an efficient 3D eddy current nondestructive 14 evaluation (ECNDE) forward solver is proposed to make estimations for PoD study. Calculation of PoD curves assisted by the proposed simulation model is performed on a 21 finite thickness plate with a rectangular surface flaw. Simulation models are proposed to deal with the required large data set. The work is interesting, and worth to be published. However, some question should be answered before publication.

1)The singular value decomposition method used in this paper is general, the authors should clarify the novelty of this work.

2)The author took too much space to clarify the memory and CPU time of the iteration. More explanation of simulation models are expected.

3)More detailed of the experiment(e.g. excitation frequency, mode of eddy current,

4)The diagram of the eddy current system used in this work should sketched and explained in detailed.

Author Response

Please find the responses in the attachement.

Reviewer 2 Report

The authors developed a novel ACA-SVD based BEM eddy current NDE solver which can calculate impedance variations and the probability of defect detection. The manuscript is well written and could be published subecting to minor revisions.

(1) Only surface flaw examples are used to verify the method. For a general NDE method, embedded flaws exist.  Does the method work for embedded flaws?

(2) For a BEM method, mesh sensitivity is important. A result or plot to show the impedance variation versus the mesh size will strengthen the manuscript.

(3) Multiple flaws are the challenge for NDEs. The proposed method will be a good contribution to the field if it can be used for such a case with the additional potential to identify flaw length, flaw size, etc.

Author Response

(The authors gave the same response as above.)

Round 2

Reviewer 1 Report

The authors addressed the question appropriately. The paper is worthed to be publised.